# Health-Related Quality of Life and Healthcare Events in Patients with Monotherapy of Anti-Diabetes Medications

**DOI:** 10.3390/healthcare11040541

**Published:** 2023-02-12

**Authors:** Tadesse Melaku Abegaz, Askal Ayalew Ali

**Affiliations:** Economic, Social and Administrative Pharmacy (ESAP), College of Pharmacy and Pharmaceutical Sciences, Institute of Public Heath, Florida A&M University, Tallahassee, FL 32307, USA

**Keywords:** anti-diabetes medications, comparative effectiveness, diabetes, quality of life

## Abstract

This study aimed to examine the difference in health-related quality of life (HRQOL) and diabetes-related healthcare events (HCEs) among adults with diabetes who were on metformin, sulfonylurea, insulin, or thiazolidinedione (TZD) monotherapy. The data were sourced from the Medical Expenditure Panel Survey (MEPS). Diabetes patients ≥18 years old who had a complete record of physical component score and mental component scores in round 2 and round 4 of the survey were included. The primary outcome was HRQOL of diabetes patients as measured by the Medical Outcome Study short-form (SF-12v2^TM^). Multinomial logistic regression and negative binomial regression were conducted to determine associated factors of HRQOL and HCE, respectively. Overall, 5387 patients were included for analysis. Nearly 60% of patients had unchanged HRQOL after the follow-up, whereas almost 15% to 20% of patients showed improvement in HRQOL. The relative risk of declined mental HRQOL was 1.5 times higher relative to unchanged mental HRQOL in patients who were on sulfonylurea 1.55 [1.1–2.17, *p* = 0.01] than metformin users. The rate of HCE decreased by a factor of 0.79, [95% CI: 0.63–0.99] in patients with no history of hypertension. Patients on sulfonylurea 1.53 [1.20–1.95, <0.01], insulin 2.00 [1.55–2.70, <0.01], and TZD 1.78 [1.23–2.58, <0.01] had increased risk of HCE compared to patients who were on metformin. In general, antidiabetic medications modestly improved HRQOL in patients with diabetes during the follow-up period. Metformin had a lower rate of HCE as compared to other medications. The selection of anti-diabetes medications should focus on HRQOL in addition to controlling glucose level.

## 1. Introduction

It is predicted that the number of patients suffering from diabetes mellitus (DM) will increase to 366 million in 2030 globally [1]. In the United States (US), DM affects more than 10% of the population [2]. DM causes multiple complications, including cardiovascular, renal, eye, and extremities. It has been revealed that DM and its complications are negatively associated with the health-related quality of life (HRQOL) of patients [3].

Pharmacotherapeutic interventions play a pivotal role in controlling high glucose, preventing diabetes-related complications, improving survival, and improving HRQOL of patients with diabetes [4,5]. Metformin, sulfonylurea, insulin, and thiazolidinediones (TZD) are the commonly used anti-diabetes medications. The American Diabetes Association (ADA) recommends these medications as the first-line therapy owing to their affordability and proved safety [6]. Previous studies in the US also reported that the use of metformin and insulin analogues increased from 1999 to 2018, derived from an ongoing safety and effectiveness study on these agents [7,8]. Despite extensive research on the glycemic effects of these first-line medications, their impact on HRQOL has yet to be explored in a real-world scenario. In addition to the aforementioned glucose-lowering agents, several hypoglycemic agents have been approved for the management of diabetes most recently. Even though the introduction of newer glucose-lowering agents such as GLP-1 agonists and SGLT2 inhibitors improves the management of DM, they are usually prescribed as a second-line therapy in the case of uncontrolled diabetes. In addition, the high cost of these medications limits their utilization for patients mainly with cardiorenal complications [9].

Few clinical trials studies have integrated patient-reported outcomes measures to evaluate the effectiveness of different pharmacotherapeutic interventions on HRQOL [10,11,12,13]. However, the generalizability of these clinical trials is limited. Meraya et al. (2020) reported poor HRQOL in patients with diabetes complications using the Medical Expenditure Panel Survey (MEPS) data [14]. A study conducted by Campbell et al. (2017) revealed that HRQOL of patients with diabetes declined as the medical expenditure of patients with diabetes increased [15]. However, these studies did not compare the effect of treatments on HRQOL and the difference in effectiveness across various anti-diabetes medications. In addition, there are limited data on diabetes-related healthcare events (HCEs), including emergency, inpatient, and outpatient visits in patients who use selected anti-diabetes medications.

The aim of this study was twofold: (1) to explore the difference in HRQOL in patients who were on monotherapy of the common anti-diabetes treatments; (2) to investigate healthcare events associated with diabetes across diabetes treatments. Investigation of the HRQOL of interventions would improve decision making during prescription by providing evidence on the comparative effectiveness of diabetes treatments.

## 2. Materials and Methods

### 2.1. Study Design and Population

A longitudinal study was conducted on non-institutionalized US civilian diabetes patients using the Medical Expenditure Panel Survey (MEPS), a nationally representative survey from 2000–2019 [16]. MEPS collects data from each participant in five consecutive rounds. Information on HRQOL is only collected in rounds 2 and 4. In each round, information on prescribed medicines is recorded including name, price, dose, and prescription dates.

Our study population included all diabetes patients above the age of 18 who participated in the MEPS survey in rounds 2 and 4, which were approximately 1 year apart. Participants who started their medication prior to round 2 were eligible to participate in the study, allowing for the measurement of changes in HRQOL between rounds 2 and 4. All eligible patients were also required to have a complete record of HRQOL. In addition, diabetes patients who received only monotherapy of metformin, sulfonylurea, insulin, or TZD were included in the study. Our exclusion criteria were patients who purchased their medication between rounds 3 and round 5, patients on combination glucose-lowering agents, patients with an incomplete record of HRQOL, and patients who changed therapy between the two rounds.

### 2.2. Primary Outcomes and Study Variables

The primary outcome of this study was HRQOL of diabetes patients who were on common antidiabetic medicines. The secondary outcome constituted HCEs, which included home health events, outpatient events, inpatient events, emergency room events, and hospitalizations associated with diabetes. The “home health events” variable in MEPS encompasses data on the use of in-home health services, including home health visits, hospice care, and other medical treatments provided in a person’s residence. This variable provides insights into the type, frequency, and cost of home health services received by individuals, enabling the analysis of trends and patterns in the utilization and spending of home health services. Inpatient events and hospitalizations include diabetes-related complications such as diabetic ketoacidosis, myocardial infarctions, stroke, foot, or urinary tract infections. All other events related to diabetes but not requiring admission to the hospital were categorized under outpatient visit such as kidney, cholesterol, eye, and hemoglobin A1C exams. The reason for all events was the diabetic condition.

The independent variables were sociodemographic characteristics of patients including sex, age, race, ethnicity, marital status, employment status, and education. Comorbidity conditions such as coronary heart disease, asthma, hypertension, angina, stroke, myocardial infarction, emphysema, arthritis, cancer, and dyslipidemia were included as additional explanatory variables.

### 2.3. Data Source and Data Collection

We used data from Medical Expenditure Panel Survey Household Component (MEPS-HC). The household component of MEPS consists of data collected from each household member on demographic characteristics, medical conditions, health status, events, prescription medicines, and medical expenses. The four classes of glucose-lowering agents were retrieved from the prescription medication file, which provides information about the therapeutic classes of medicines in connection with the Multum Lexicon database. The diagnosis of DM and other conditions was identified from the medical condition file.

The HRQOL was measured using the Medical Outcome Study 12 Item Short-Form (SF-12v2^TM^), a standardized generic questionnaire of medical outcome study which contains 12 items. Since 2000, SF-12v2^TM^ has been administered to adults above 18 years old. The SF-12v2 was validated using MEPS for measuring HLQOL in diabetic patients in the US (Cronbach α: PCS = 0.85; MCS = 0.83) [17]. It was also validated to measure HRQOL by Ware et al. in the general population [18]. The eight domains of SF-12v2^TM^ are summarized into two components: the physical component score (PCS) and the mental component score (MCS). The MCS and PCS are scored out of 100. Higher scores are associated with better physical and mental health. An average score of both MCS and PCS is 50 points for the general US population [18]. For the purpose of this study, classification of the HRQOL status was carried out using the minimum clinically important difference (MCID) method, which is described below.

### 2.4. Minimum Clinically Important Difference in HRQOL

We computed the MCID to determine whether the change in HRQOL between round 2 (before treatment) and round 4 (after treatment) was clinically meaningful [19]. MCID can be computed using different approaches. The distribution-based approach is the common method that estimates MCID using different measures. The half standard deviation (half-SD) is utilized as a common distribution-based method. According to previous studies, a value of half-SD approximates the threshold of discrimination for clinically meaningful changes in HRQOL for chronic diseases [20,21]. The half-SD is estimated by calculating the SD of the change in MCS and PCS scores between the two rounds. Then, HRQOL is categorized into improved, unchanged, and declined on the basis of the values of the half-SD.

### 2.5. Data Analyses

Data were cleaned and analyzed using Stata Version 15 [22]. All analyses were conducted using survey procedures. The present study considers the sampling weights (longitudinal weights), clustering, and stratification design to determine HRQOL and diabetes-related events. Descriptive statistics were performed on the patient’s demographic and clinical characteristics, while multinomial logistic regression was conducted to determine association between the different classes of medications and HRQOL. The categorical forms of the mental and physical components of HRQOL were our dependent variables. Negative binomial regression was conducted to assess factors associated with HCE. HCE was a count variable with unequal mean and variance. The negative binomial model was the best fit model for such data. A *p*-value < 0.05 was set a priori with a 95% confidence interval to test the level of significance.

## 3. Results

### 3.1. Characteristics of Diabetes Patients

Overall, 5387 individuals met the inclusion criteria (weighted estimate: 103,169,500). Overall, 50% of participants were males (49.33%). The mean age of participants was 60.65 (SD: 20.78). More than three-fourths of the participants were non-Hispanic whites (73.53%). The majority were married (58.25%). Almost 19% of patients had health insurance. About 40% of them were employed. Approximately 11% had stroke, 12% had asthma, and one-half of the participants (49.92%) had arthritis. More than 70% of the population had dyslipidemia (73.18%). Three-fourths (75.15%) of the subjects had high blood pressure. There was a significant difference among participants in the four medication groups in terms of sex, race, marital status, ethnicity, age as a categorical variable, stroke, angina, high cholesterol, arthritis, and diabetes-related eye complications (*p* < 0.01), whereas no significant differences were reported in other characteristics, including education, health insurance, coronary heart disease, high blood pressure, emphysema, and asthma (Table 1).

### 3.2. Health-Related Quality of Life of Diabetes Patients

More than 60% of patients had unchanged status in mental-HRQOL (62.91%) and physical-HRQOL (67.02%) components, whereas almost 15% to 20% of patients showed an improvement in HRQOL for all monotherapy users as compared to the baseline (Table 2). The maximum improvement in HRQOL among individual antidiabetic medications did not exceed 5% between the two periods, and the HRQOL improvement was comparable across medications. In the metformin group, 4.76% and 4.1% of individuals had improved in MCS and PCS components, respectively. The improvement in HRQOL in the insulin group ranged from 4.7% on MCS and 4.78% on PCS. In the sulfonylurea group, the improvement in HRQOL was between 4.57% and 5.59% (Table 2).

The overall trend of HRQOL increased slightly in the MCS component across different panels with an overall mean ± SD of 45.55 ± 15.77 (interquartile range = 38.51–57.16). A sharp decline in MCS was observed in panel 19 (mean MCS = 40.652) (Table 2). The trend of the PCS score remained stable across different panels (mean ± SD = 37.679 ± 15.19, interquartile range = 27.751–50.22) (Figure 1 and Figure 2).

### 3.3. Factors Associated with Health-Related Quality of Life of Diabetes Patients

To determine factors associated with HROQL, we ran a multinomial logistic regression. It was found that, among many factors, age, race, and antidiabetic medications were found to be associated with HRQOL. The relative risk for improved mental HRQOL status relative to unchanged status increased by 50% in patients aged 40 to 49 and 60 to 69 years old (0.53 [95 CI% 0.27–0.92, *p* = 0.02] and 0.50 [95 CI% 0.32–0.98, *p* = 0.02], respectively), as compared with patients aged between 18 and 39. Relative to Hispanics, the relative risk of improved mental HRQOL relative to unchanged mental HRQOL was 48% lower for Blacks 0.52 [0.34–0.79, <0.01] and 55% lower for NH whites 0.45 [0.28–0.73, <0.01]. The relative risk of declined mental HRQOL as compared to unchanged mental HRQOL increased by 1.45 in patients with a history of unemployment 1.45 [1.02–2.04, 0.03] as compared with employed individuals. The relative risk of declined mental HRQOL was 1.5 times higher relative to unchanged mental HRQOL in patients who were on sulfonylurea than metformin users 1.55 [1.1–2.17, *p* = 0.01]. In comparison to Hispanics, non-Hispanic whites had 40% less risk of improved physical HRQOL 0.60 [0.41–0.90, *p* = 0.01] relative to unchanged physical HRQOL. In contrast to the Hispanic population, Black patients had about 37% lower risk of improved physical HRQOL relative to unchanged physical HRQOL (0.63 [0.41–0.96, *p* = 0.03]). Patients with the age range of 50–59 had almost 45% lower risk of improved physical HRQOL relative to unchanged physical HRQOL (0.55 [0.31–0.96, *p* = 0.04]; Table 3 and Table 4).

### 3.4. Healthcare-Associated Events in Diabetes Patients

One-quarter of patients had no healthcare-associated events (23.46%). About one-fifth (18.85%) of patients had greater than six events. Around 18.2% of patients had at least one HCE. There was a significant difference among participants who received different anti-diabetes treatments in terms of the number of HCEs associated with diabetes, including emergency visits, home health visits, inpatient events, and office-based events (*p* < 0.01). No significant difference was observed between different treatment groups regarding hospital stay and outpatient events (Table 5).

### 3.5. Determinants of Healthcare Events Associated with Diabetes

According to the result of the negative binomial regression model, the most significant determinant factors associated with the occurrence of HCEs are shown in Table 6. Compared with married individuals, being widowed increased the risk of HCEs by a factor of 1.46 [1.02–2.1, *p* < 0.01]. Relative to employed individuals, patients without employment had 1.44 times greater rate for HCE [1.16–1.79, *p* = 0.01]. The rate of HCEs decreased by a factor of 0.79 in patients without a history of hypertension compared hypertensive patients [0.63–0.99, *p* = 0.04]. Patients on sulfonylurea (1.53 [1.20–1.95, <0.01]), insulin (2.00 [1.55–2.70, <0.01]), and TZD (1.78 [1.23–2.58, <0.01]) had increased risk of HCEs compared to patients who were on metformin (Table 6).

## 4. Discussion

Most studies have focused on the evaluation of intermediate clinical endpoints of diabetes (i.e., glycemic control) following anti-diabetes treatments. Recently, a few comparative effectiveness studies integrated patient reported outcomes measures along with randomized control trial studies to measure HRQOL [10,11,12,13]. However, a real-world evaluation of HRQOL of diabetes patients on different treatment modalities has not been sufficiently explored [23]. This study aimed to evaluate the HRQOL and HCE of patients with diabetes who were taking selected anti-diabetes treatments.

According to our findings, about 15–20% of patients showed an improvement in HRQOL from the baseline HRQOL level. More than 20% of the population had unchanged HRQOL at the end of the follow-up. The trend in HRQOL did not show a significant change across different panels in physical HERQOL, but a sharp decline was observed in mental HRQOL from panels 18 to 19 before increasing afterward. Several factors were associated with poor HRQOL, including history of unemployment, type of anti-diabetes, Black and white races, and age ranges of 40–49 and 60–69. More than 20% of diabetes individuals experienced at least one HCE associated with diabetes. Multiple factors influenced the occurrence of HCEs in diabetes patients such as marital status, employment status, and anti-diabetes medications.

Our study indicated that the use of anti-diabetes medications modestly improved HRQOL in diabetes patients. This could be through achieving adequate glycemic control [24,25]. The effectiveness of these medications to improve HRQOL could also be attributed to preventing diabetes-related complications [26,27]. For instance, TZDs were reported to confer cardioprotective effect. According to recent studies, TZDs such as pioglitazone significantly improve endothelial and adipose tissue dysfunction and reduce the composite of nonfatal myocardial infarction and stroke in patients with type 2 diabetes, which might contribute to HRQOL improvement [28,29]. However, there is controversy about the role of TZD on CVD outcomes as they were once labeled as a black-box warning for worsening of heart failure in in diabetes patients [30]. On the other hand, evidence on the effects of sulfonylureas on the heart is still not conclusive and remains an ongoing debate [31]. The modest improvement in HRQOL associated with the use of these glucose-lowering agents might also be ascribed to the use of a single blood glucose-lowering agent. Rizza et al. (2021) noted that using a single dose of anti-diabetes medication resulted in a greater likelihood of improving health and quality of life as compared to combination therapies [32]. Additional pharmacotherapeutic properties of individual anti-diabetes medications could also contribute to their impact on HRQOL. For example, metformin decreases obesity, which might affect physical health. However, some safety profiles, such as undesired weight gain due to insulin and sulfonylurea might attenuate their benefit on HRQOL [33,34]. Weight gain has a negative psychological impact and compromises physical strength, which might expose one to frailty [35]. As noted in the multinomial regression report, a relatively higher rate of decline in HRQOL associated with sulfonylureas and TZD as compared to metformin could be attributed to safety differences between treatments [36]. However, controlled studies with an adequate follow-up period are required to capture the difference in HRQOL between these agents.

The current study explored the magnitude of improvement in HRQOL as influenced by race and ethnicity. Black Americans and NH whites had a higher rate of decline in HRQOL. The disparity in HRQOL might be due to the variation in response to anti-diabetes medications between different racial/ethnic groups and the difference in socioeconomic status [37]. Even though patients had equal probability of taking anti-diabetes medications, the difference in economic status could affect access to psychological services, especially for African-Americans [38]. Despite the socioeconomic disadvantages surrounding the Hispanic population, Hispanics in the United States tend to have significantly better health outcomes than the average population, which was also observed in our study [39]. The present study also reported variation in the HRQOL between different age groups. It was found that middle-aged to older individuals tended to have lower HRQOL than young patients [40]. As age increased, the number of comorbidities and healthcare events become more pronounced, resulting in decreased HRQOL [41]. On the other hand, some studies reported that HRQOL was better in older patients, which might be related to the high amount of healthcare received by the older population [38,39,40,41,42].

In the present study, one in five individuals had at least one HCE associated with diabetes. As with many chronic conditions, diabetes is known to cause multiple inpatient and outpatient events [43]. A number of studies reported that patients with diabetes had a 2–6 times higher rate of admission than patients without diabetes [44,45]. In 2015, it was estimated that 92 in 1000 diabetes patients would visit an emergency room in the US [46]. HCEs related to diabetes can occur at the onset of the disease in the form of diabetic ketoacidosis or micro- and macrovascular complications. The frequent precipitation of hyperglycemia and the progression of the disease add a tally to several admissions [47]. The occurrence of these events might vary according to the type of pharmacologic management. In our study, it was stated that the use of sulfonylureas, insulin, and TZD was likely to increase the rate of HCEs as compared to metformin. The high rate of HCEs in these medications could be inadequate glucose control when given as monotherapy [48]. Poor insulin administration might also exacerbate hyperglycemia and emergency visits [44]. Roumie et al. (2012) reported that the use of sulfonylureas for the treatment of diabetes was associated with an increased hazard of diabetes-related events (18.2 per 1000 person-years) compared to metformin users (10.4 per 1000 person-years) [49]. Lipscombe et al. (2007) reported that treatment with TZD monotherapy was associated with a significantly increased risk of congestive heart failure, acute myocardial infarction, and death compared with other oral hypoglycemic therapies that led to emergency room visits [50]. Thus, HCEs associated with diabetes could be mitigated with the appropriate selection of anti-diabetes medications.

In general, the present study provided important information on HRQOL and HCEs in patients with diabetes who were on different anti-diabetes medications. The representativeness of the sample could allow generalizability of the findings. However, the retrospective design of the study did not enable to control all confounders of HRQOL. In addition, the duration of therapy between the two timepoints may not have been able to capture significant differences in HRQOL between the two rounds. The HRQOL assessment tool is not specific to diabetes, which could also have affected our estimation of HRQOL in these special population. The study was also affected by bias from self-reported HRQOL. Our study did not incorporate the impact of nonpharmacological interventions that might influence HRQOL. The current study did not evaluate the difference in HRQOL of other anti-diabetes medications such as sodium glucose transporter inhibiters and incretin mimetics that are usually combined with one or more of selected anti-diabetes medications. The findings of the study should be interpreted in light of these limitations.

## 5. Conclusions and Recommendations

In conclusion, antidiabetic medications modestly improved HRQOL in patients with diabetes during the follow-up period. The overall improvement in HRQOL was approximately 20%. Patients with sulfonylurea had a higher decline in HRQOL. Patients on metformin had a lower rate of HCEs as compared to other medications. There was a difference in HRQOL in terms of race and age. The selection of anti-diabetes medication should focus on HRQOL in addition to intermediate outcomes (i.e., glucose level). Special attention is required for patients with various age groups and ethnic origins to improve HRQOL in patients with diabetes. Policymakers should ensure the availability of glucose-lowering agents that demonstrate high yield of HRQOL at affordable price. The enrolment of diabetes patients to different health plans such as traditional Medicare versus Medicare advantage could cause variation in the uptake of anti-diabetes medications [51]. Efforts should be sought to eliminate disparity in receiving these glucose-lowering agents across diabetes patients who are enrolled in different health plans. Engagement of patients’ view during drug selection could also help to initiate appropriate anti-diabetes medication with better HRQOL. Future research can be directed to comparing the HRQOL of patients taking combination therapies with respect to other new therapeutic alternatives such as GLP-1 agonists and SGLT2 inhibitors.

## Figures and Tables

**Figure 1 healthcare-11-00541-f001:**
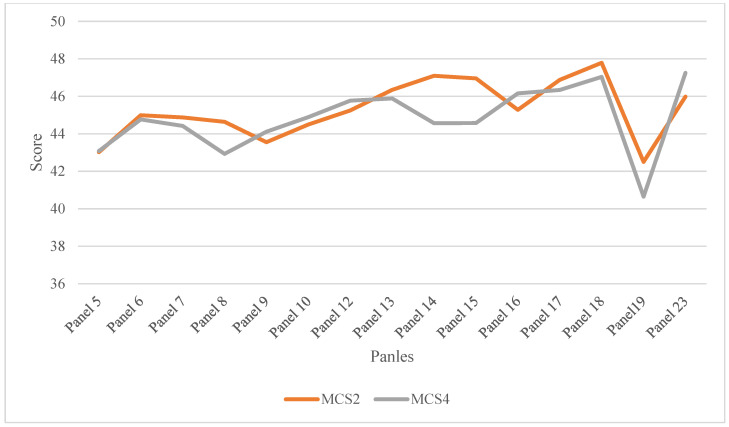
The trend of mental component score (MCS) domain: Panels 11, 20, 21, and 22 are not shown in the graph due to an incomplete record of HRQOL. MCS2: mental component score in round 2, MCS4: mental component score in round 4.

**Figure 2 healthcare-11-00541-f002:**
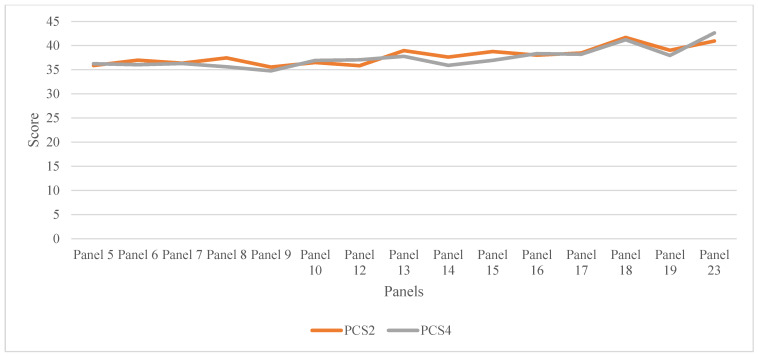
The trend of physical component score (PCS) domain, panels 5–23. Panels 11, 20, 21, and 22 are not shown in the graph due to an incomplete record of HRQOL. PCS2: physical component score in round 2, PCS4: physical component score in round 4.

**Table 1 healthcare-11-00541-t001:** Characteristics of the study participants, Wt (%).

Variables	Metformin	Sulfonylurea	Insulin	TZD	Total	*p*-Value
Sex						
Male	13.65	16.51	10.57	8.6	49.33	0.02
Female	16.24	15.11	10.55	8.76	50.67	
Age (mean ± SD)	58.68 (32.85)	63.3 (31)	58.44 (45.45)	61.87 (34.6)	60.65 (20.78)	0.5144
18–39	2.42	1.4	2.78	0.56	7.16	<0.01
40–49	4.91	3.65	2.94	2.03	13.52	
50–59	7.97	6.81	4.31	4.58	23.67	
60–69	7.91	8.81	5.27	5.59	27.57	
70+	36.69	10.95	5.83	4.61	28.07	
Race						
White	21.99	22.83	14.31	14.39	73.53	
Black	4.46	4.66	4.37	2.45	15.93	<0.01
American Indian	0.35	0.53	0.27	0.23	1.38	
Asian	1.36	1.69	0.64	0.63	4.33	
Native Hawaiian/Pacific Islander	0.83	1.42	0.71	0.3	3.25	
Others	0.49	0.43	0.3	0.36	1.58	
Ethnicity						
Hispanic	4.41	5.05	2.57	2.39	14.42	
NH Black	8.99	8.49	7.19	3.11	27.79	<0.01
NH Asian	2.7	3.44	1.91	0.96	9	
NH white	13.5	14.32	9.29	10.85	47.95	
NH other	0.29	0.32	0.17	0.05	0.84	
Marital status						
Married	16.52	18.6	11.65	11.48	58.25	0.04
Widowed	4.02	5.56	3.15	3.04	15.74	
Divorced	3.91	4.18	2.82	2.62	13.53	
Separated	0.6	0.7	0.64	0.53	2.48	
Never married	2.76	2.31	2.73	2.19	2.19	
Education						
No degree	6.75	7.67	4.56	3.66	22.64	0.707
High school diploma	15.09	16.82	11.81	8.65	52.37	
Bachelor’s degree and above	7.57	7.26	5.45	4.71	24.99	
Health insurance status (Yes)	27.77	28.71	17.89	14.21	88.57	0.496
Comorbidity						
Coronary heart disease	3.69	5.65	5.18	3.15	0.68	0.68
Asthma	4.05	3.82	3.15	1.5	12.52	0.612
Hypertension	21.7	24.68	15.6	13.17	75.15	0.194
Angina	2.13	2.55	1.89	9.35	15.92	<0.01
Myocardial infarction	3.14	4.99	3.89	2.16	14.17	<0.01
Stroke	2.5	3.65	3.3	1.69	11.14	<0.01
Emphysema	1.34	1.76	0.9	0.61	4.61	0.417
Arthritis	14.41	15.81	10.51	9.19	49.92	0.106
Dyslipidaemia	21.87	23.29	15.02	13	73.18	<0.01
Cancer	5.00	6.24	5.05	2.77	19.06	0.042
Employment (yes)	15.31	11.53	8.27	7.71	42.83	<0.01

Wt: weighted, TZD: thiazolidinedione, NH: non-Hispanic, SD: standard deviation.

**Table 2 healthcare-11-00541-t002:** Health-related quality of life of diabetes patients.

Treatments	MCS Wt (%)			PCS Wt (%)		
	Improved	Unchanged	Declined	Improved	Unchanged	Declined
Metformin	4.76	19.86	5.27	4.1	20.53	5.26
Sulfonylureas	5.59	19.46	6.58	4.57	20.8	6.25
Insulin	3.7	12.51	4.47	3.78	13.51	3.84
TZD	2.82	11.09	3.46	2.22	12.18	2.94
Total	20.23	62.91	16.86	14.7	67.02	18.28

MCS: mental component score, PCS: physical component score, TZD: thiazolidinedione, Wt: weighted.

**Table 3 healthcare-11-00541-t003:** Factors affecting mental HRQOL of diabetes patients.

Variables	Declined		Improved	
	RRR 95% CI	*p*-Value	RRR 95% CI	*p*-Value
Sex (Ref: male)				
Female	1.13 [0.8430–1.53]	0.4	1.22 [0.88–1.69]	0.22
Age (Ref: 18–39)				
40–49	0.62 [0.33–1.18]	0.14	0.53 [0.27–0.92]	0.02
50–59	0.92 [0.51–1.65]	0.78	0.74 [0.44–1.24]	0.25
60–69	0.77 [0.42–1.42]	0.4	0.50 [0.32–0.98]	0.02
70+	0.81 [0.45–1.44]	0.47	0.76 [0.43–1.35]	0.35
Race (Ref: Hispanic)				
NH Black	0.82 [0.56–1.2]	0.31	0.52 [0.34–0.79]	<0.01
NH Asian	0.95 [0.61–1.49]	0.82	0.99 [0.64–1.52]	0.96
NH white	0.87 [0.55–1.38]	0.56	0.45 [0.28–0.73]	<0.01
NH others	1.16 [0.46–2.92]	0.46	1.17 [0.38–3.59]	0.78
Insurance (Ref: yes)				
No	1.31 [0.83–2.05]	0.24	1.1 [0.65–1.8]	0.76
Employment (Ref: yes)				
No	1.45 [1.02–2.04]	0.03	1.08 [0.77–1.5]	0.8
Coronary artery disease (Ref: yes)				
No	1.032 [0.63–1.69]	0.89	0.86 [0.54–1.36]	0.51
Asthma (Ref: yes)				
No	0.81 [0.54–1.19]	0.28	1.22 [0.77–1.92]	0.39
Hypertension (Ref: yes)				
No	0.88 [0.61–1.27]	0.5	0.94 [0.63–1.4]	0.78
Angina (Ref: yes)				
No	0.85 [0.48–1.48]	0.56	0.93 [0.54–1.58]	0.78
Myocardial infarction (Ref: yes)				
No	0.82 [0.45–1.47]	0.51	1.13 [0.66–1.93]	0.64
Emphysema (Ref: yes)				
No	0.61 [0.31–1.18]	0.14	0.64 [0.29–1.43]	0.27
Dyslipidaemia (Ref: yes)				
No	1.32 [0.89–1.97]	0.16	0.85 [0.57–1.25]	0.41
Cancer (Ref: yes)				
No	0.76 [0.54–1.05]	0.1	1.14 [0.76–1.71]	0.51
Arthritis (Ref: yes)				
No	0.83 [0.58–1.17]	0.29	0.82 [0.56–1.19]	0.3
Healthcare events (Ref: 0)				
One events	0.84 [0.51–1.39]	0.4	0.76 [0.49–1.18]	0.22
2–3 events	0.8 [0.52–1.23]	0.32	1.11 [0.74–1.66]	0.60
4–5 events	0.79 [0.43–1.46]	0.46	1.08 [0.6–1.96]	0.77
≥6 events	0.89 [0.55–1.42]	0.62	0.91 [0.55–1.5]	0.72
Anti-diabetes (Ref: metformin)				
Sulfonylureas	1.55 [1.1–2.17]	0.01	1.11 [0.72–1.73]	0.62
Insulin	1.39 [0.88–2.22]	0.15	1.09 [0.70–1.68]	0.70
TZD	1.69 [1–2.89]	0.05	0.94 [0.55–1.6]	0.55

NH: Non-Hispanic, RRR: relative risk ratio, TZD: thiazolidinedione.

**Table 4 healthcare-11-00541-t004:** Factors affecting physical HRQOL of diabetes patients.

Variables	Declined		Improved	
	RRR 95% CI	*p*-Value	RRR 95% CI	*p*-Value
Sex (Ref: male)				
Female	0.93 [0.69–1.28]	0.69	1.35 [0.9–1.9]	0.06
Age (Ref: 18–39)				
40–49	0.73 [0.38–1.4]	0.35	0.80 [0.42–1.5]	0.50
50–59	0.81 [0.47–1.4]	0.44	0.55 [0.31–0.96]	0.04
60–69	0.72 [0.42–1.2]	0.22	0.84 [0.48–1.47]	0.55
70+	1.06 [0.38–1.4]	0.9	0.70 [0.39–1.27]	0.25
Race (Ref: Hispanic)				
NH Black	1.04 [0.71–1.54]	0.82	0.63 [0.41–0.96]	0.03
NH Asian	0.81 [0.53–1.3]	0.36	0.98 [0.64–1.48]	0.9
NH white	0.78 [0.51–1.19]	0.25	0.60 [0.41–0.90]	0.01
NH others	1.16 [0.39–2.8]	0.46	0.93 [0.41–2.08]	0.85
Insurance (Ref: yes)				
No	1.19 [0.69–2.06]	0.52	1.39 [0.93–2.11]	0.11
Employment (Ref: yes)				
No	1.42 [0.94–2.15]	0.09	1.02 [0.69–1.3]	0.64
Coronary artery disease (Ref: yes)				
No	0.7 [0.46–1.1]	0.11	1.1 [0.70–1.75]	0.7
Asthma (Ref: yes)				
No	1.1 [0.74–1.58]	0.69	0.86 [0.52–1.82]	0.55
Angina (Ref: yes)				
No	1.1 [0.64–1.9]	0.72	0.65 [0.40–1.03]	0.07
Myocardial infarction (Ref: yes)				
No	0.83 [0.50–1.38]	0.48	0.74 [0.42–1.3]	0.30
Emphysema (Ref: yes)				
No	0.67 [0.36–1.22]	0.19	0.85 [0.46–1.56]	0.6
Dyslipidaemia (Ref: yes)				
No	0.87 [0.57–1.33]	0.49	1 [0.66–1.25]	0.98
Cancer (Ref: yes)				
No	1.08 [0.54–1.05]	0.8	0.9 [0.57–1.1]	0.63
Healthcare events (Ref: 0)				
One events	0.7 [0.47–1.05]	0.09	0.81 [0.50–1.13]	0.39
2–3 events	0.65 [0.42–1]	0.05	0.81 [0.52–1.4]	0.36
4–5 events	0.62 [0.35–1.08]	0.09	0.90 [0.48–1.7]	0.72
≥6 events	0.73 [0.43–1.24]	0.24	0.82 [0.49–1.37]	0.44
Anti-diabetes (Ref: metformin)				
Sulfonylureas	1.31 [0.88–1.94]	0.17	0.89 [0.62–1.28]	0.53
Insulin	0.98 [0.59–1.6]	0.92	0.96 [0.65–1.41]	0.84
TZD	1.04 [0.62–1.74]	0.89	0.78 [0.43–1.42]	0.43

RRR: relative risk ratio, TZD: thiazolidinedione.

**Table 5 healthcare-11-00541-t005:** The mean number of healthcare-associated events in diabetes patients across different treatments.

Type of Events	Metformin	Sulfonylurea	Insulin	TZD	Total	*p*-Value
Emergency events	0.03	0.03	0.09	0.02	0.17	<0.01
Home health events	0.11	0.15	0.27	0.16	0.69	<0.01
Hospital stays	0	0.03	0	0.18	0.21	0.18
Inpatient events	0.01	0.02	0.08	0.01	0.12	<0.01
Office-based events	2.24	2.86	3.55	3.51	12.16	<0.01
Outpatient events	0.13	0.16	0.23	0.27	0.79	0.08
Number of HCEs						
No event	8.89	7.5	4.18	2.88	23.46	<0.01
One events	6.24	5.72	3.49	2.76	18.2	
2–3 events	7.59	7.99	5.06	4.59	25.23	
4–5 events	3.44	4.86	2.86	3.12	14.26	
≥6 events	3.73	5.55	5.54	4.03	18.85	

HCEs: healthcare events, TZD: thiazolidinedione.

**Table 6 healthcare-11-00541-t006:** Determinants of healthcare events associated with diabetes: negative binomial regression model.

Factors	IRR 95%CI	*p*-Value
Sex (Ref: male)		
Female	0.98 [0.81–1.2]	0.88
Age (Ref: 18–39)		
40–49	1.32 [0.89–1.94]	0.16
50–59	1.13 [0.79–1.61]	0.5
60–69	0.96 [0.66–1.34]	0.82
+70	0.78 [0.55–1.13]	0.19
Race (Ref: Hispanic)		
NH Black	0.94 [0.77–1.14]	0.53
NH Asian	0.83 [0.62–1.1]	0.5
NH white	1.12 [0.75–1.66]	0.57
NH others	0.74 [0.44–1.2]	0.27
Marital status (Ref: married)		
Widowed	1.46 [1.02–2.1]	0.04
Divorced	1.14 [0.85–1.53]	0.37
Separated	0.95 [0.6–1.48]	0.81
Never married	0.99 [0.75–1.32]	0.94
Insurance (Ref: yes)		
No	1.33 [0.96–1.85]	0.08
Employment (Ref: yes)		
No	1.44 [1.16–1.79]	0.01
Coronary artery disease (Ref: yes)		
No	1.03 [0.72–1.47]	0.88
Asthma		
No	1.29 [0.94–1.79]	0.11
Hypertension (Ref: yes)		
No	0.79 [0.63–0.99]	0.04
Angina (Ref: yes)		
No	0.92 [0.61–1.38]	0.68
Stroke (Ref: yes)		
No	1.12 [0.87–1.44]	0.4
Dyslipidaemia (Ref: yes)		
No	1.19 [0.91–1.55]	0.19
Cancer (Ref: yes)		
No	1.2 [0.94–1.54]	0.15
Arthritis (Ref: yes)		
No	0.80 [0.67–0.96]	0.02
Anti-diabetes (Ref: metformin)		
Sulfonylureas	1.53 [1.20–1.95]	<0.01
Insulin	2.00 [1.55–2.70]	<0.01
TZD	1.78 [1.23–2.58]	<0.01

IRR: incidence rate ratio, NH: non-Hispanic, TZD: thiazolidinedione.

## Data Availability

Not applicable.

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
