# Peer review of "Health-Related Quality of Life and Healthcare Events in Patients with Monotherapy of Anti-Diabetes Medications"

_healthcare, 2023, doi:10.3390/healthcare11040541_

Round 1

Reviewer 1 Report

Please write “Anti-diabetes Medications” in the keywords.

Introduction

1. In line 31, please write “diabetes mellitus” and then use (DM).

2. In line 37, please use the complete words of HRQOL and then use its abbreviation for the first time.

3. Ass you are using % to present the numbers it is better to use it for all the numbers. Correct ten percent of in line 32, fifty percent in line 124…..

Materials and Methods

1. In the “Study Design and Population”, please write the year of the study or collecting data.

2. Please include all the inclusion criteria. Please write the exclusion criteria too.

3. Please write OPE instead of OP when you used outpatient events in the line 69.

4. Can you use a reference after writing “a nationally representative survey” in line 59?

5. Can you please provide some information about the stage 1 and 2 and why those who were in MEPS survey in rounds 2 and 4 were included.

6. It will be helpful the study variables also be mentioned in the methodology section.

7. When you talk about The HRQOL, it is helpful if you please talk about scoring different components of this tool.

8. Please write the level of significance in the data analysis section.

Results

1. Please add “race” as one of the significant factors when you explain the results of table 1.

2. Please move Table 2, when you talked about it in text and then write the sentence you wrote about the results of Figure 1 and 2 and then use these two figures.

3. Please write 18.2% instead of “twenty percent” in line 193.

4. It seems “Hospital Stays” is not significant in table 5. Please correct the sentences you wrote before this Table.

Conclusion

Please provide some recommendation for stakeholders, patients or future researchers.

Author Response

Dear Reviewer,

Thank you so much for your valuable comments. We have responded to your comments point by point as follows. 

Comments and Suggestions for Authors

Please write “Anti-diabetes Medications” in the keywords.

We included “Anti-diabetes Medications” in the keywords.

Introduction

  1. In line 31, please write “diabetes mellitus” and then use (DM).

We included diabetes mellitus (DM) in line 31

  1. In line 37, please use the complete words of HRQOL and then use its abbreviation for the first time.

The complete word of health-related quality of life followed by the abbreviation, (HRQOL) was stated in line 37.

  1. Ass you are using % to present the numbers it is better to use it for all the numbers. Correct ten percent of in line 32, fifty percent in line 124…..

Ten percent in line 32 and fifty percent in line 124 were changed to 10% and 50% respectively. In addition, all percentages were changed to % to make the use consistent throughout the manuscript.

Materials and Methods

  1. In the “Study Design and Population”, please write the year of the study or collecting data.

We revised the statement in line 57-59 as follows.

“A longitudinal study was conducted on non-institutionalized U.S civilian diabetes patients in the US between 2000-2019, using the Medical Expenditure Panel Survey (MEPS), a nationally representative survey”. 

  1. Please include all the inclusion criteria. Please write the exclusion criteria too.

The inclusion and exclusion criteria were described as follows.

“Eligible participants were all diabetes patients above the age of 18 who participated in MEPS survey round 2 and round 4 which are approximately one year apart. Information on HRQOL is collected only in rounds 2 and 4 from each participant. Participants should purchase their medications in the first and second rounds for included in the study. All eligible patients were also required to have a complete record of HRQOL. In addition, diabetes patients who only received monotherapy of metformin, sulfonylurea, insulin, and TZD were included in the study. We excluded patients who started anti-diabetes medications between round 3 and round 5, patients on combination glucose-lowering agents, patients with an incomplete record of HRQOL in the two rounds, and patients who changed therapy between the two rounds.”

  1. Please write OPE instead of OP when you used outpatient events in the line 69.

We removed “OP” from the list of abbreviations since we did not find the abbreviation relevant.  therefore, we did not require to make this change.

  1. Can you use a reference after writing “a nationally representative survey” in line 59?

We cited the following reference next to “a nationally representative survey”

Agency for Healthcare Research and Quality (AHRQ). Medical expenditure panel survey, Survey Background. Available from: https://www.meps.ahrq.gov/mepsweb/about_meps/survey_back.jsp. Accessed 31 Jan 2023.

  1. Can you please provide some information about the stage 1 and 2 and why those who were in MEPS survey in rounds 2 and 4 were included.

The inclusion of the MEPS survey only in rounds 2 and 4 is due to the fact that MEPS collects data on HRQOL from each participant. In every round, information on prescribed medicines is recorded, including the name, price, dose, and prescription dates. Participants who started their medication prior to round 2 are eligible to participate in the study, allowing for the measurement of changes in HRQOL between rounds 2 and 4. 

  1. It will be helpful the study variables also be mentioned in the methodology section.

The dependent variables were HRQOL and HCE. The independent variables include Sociodemographic characteristics of patients such as sex, age, race, ethnicity, marital status, employment status and education. Comorbid conditions such as coronary heart disease, asthma, hypertension, angina, stroke, myocardial infarction, emphysema, arthritis, cancer, and dyslipidemia were also included as additional explanatory variables. This information is added into the methods section.

  1. When you talk about the HRQOL, it is helpful if you please talk about scoring different components of this tool.

The following text is added to explain the difference in the scoring difference of HRQOL domains: The MCS and PCS are scored out of 100. The higher scores are associated with better physical and mental health. An average score of both MCS and PCS is 50 points for the general US population. We included this additional information in the manuscript.

  1. Please write the level of significance in the data analysis section.

We included this information in the manuscript.  

“A p-value of <0.05 was set at priori with 95% confidence interval to test the level of significance.”

Results

  1. Please add “race” as one of the significant factors when you explain the results of table 1.

Race is added as a significant factor under the description of characteristics of diabetes patients in Table 1.

  1. Please move Table 2, when you talked about it in text and then write the sentence you wrote about the results of Figure 1 and 2 and then use these two figures.

Table 2 was moved under the text that talks about Health-Related Quality of Life of Diabetes Patients while the figures were placed following the text that describes trend in MCS and PCS components.

  1. Please write 18.2% instead of “twenty percent” in line 193.

18.2% was written instead of 20%

  1. It seems “Hospital Stays” is not significant in table 5. Please correct the sentences you wrote before this Table.

We have corrected this statement by specifying significant and non-significant values.

Conclusion

Please provide some recommendation for stakeholders, patients or future researchers.

Policymakers should ensure the availability of glucose-lowering agents that demonstrate high yield of HRQOL at affordable prices. Engaging patients’ views during clinical decision-making could also help to initiate appropriate anti-diabetes medication with better HRQOL. Future research can be directed to compare the HRQOL of patients who are taking combination therapies or with respect to other new therapeutic alternatives such as GLP-1 agonists and SGLT2 inhibitors. 

Reviewer 2 Report

If the authors discussion about the use of traditional Medicare and the its role for anti-diabetes it will nice.

Author Response

Thank you so much for your valuable comments. We have responded to your comments point by point as follows. 

  1. If the authors discussion about the use of traditional Medicare and its role for anti-diabetes it will nice.

This statement was included in the conclusion section.

The enrolment of diabetes patients in different health plans such as traditional Medicare versus Medicare advantage could cause variation in the uptake of anti-diabetes medications [46]. Efforts should be sought to eliminate of disparity in receiving these glucose-lowering agents across diabetes patients who are enrolled in different health plans.

Reviewer 3 Report

In this manuscript the authors examine the difference in health-related quality of life and diabetes-related healthcare events in a large group of adults with diabetes taking a monotherapy including Metformin, Sulfonylurea, Insulin, or Thiazolidinediones (TZD). They find that antidiabetic medications modestly improved QOL during the follow-up period, suggesting that the selection of anti-diabetes medications should be done taking into account also QOL of diabetic patients. The stuy is interesting. However, it deserves to be deeply improved.

Maior concerns:

1) Authors appear to be particularly careful and enthusiasts to provide a long list of acronyms. They occupy larger space of the manuscript for the description of several acronyms rather than try to explain the main findings of this study. Actually, throughout the text this reviewer counts more than 20 acronyms that are really too many. In particular, to read and understand "data source and data collection" as well as "minimun clinically important defference in HRQOL" is difficult and annoying. This part of the manuscript urgently needs to be semplify.

2) When does this survey begin? When does it end?

3) Please clarify what authors indicate for "home health events"; falls, household accidents, burns or whatelse? What outpatients events include?, hyperglicemia?, hypertensive peaks?; and what includes inpatients events? stroke, myocardial infarction, or nosocomial infection??

4) In introduction, authors must better explain why they opted to explore the effects of 4 classes of diabetic drugs belonging to an old scientific era. Actually, the recent european and american guidelines for treatment of diabetes indicate SGLT-2-i and GLP1-AR as the first line classes to administer to a diabetic patients.

5) In Discussion, authors must better distinguish between the different effect of anti-diabetic drugs on outcomes. TZDs (such as pioglitazone) have evident metabolic (doi: 10.1016/j.atherosclerosis.2010.12.021) and cardioprotective effects in patients in secondary CV prevention (PROACTIVE study). Please include these 2 studies. Furthermore, please edit lines 231-233 and lines 236-237. Moreover, the contoversial effect of sulpholinireas on CAD must be better discussed

Minor concers

1) Lipid dyslipidemia? do authors by any chance know any kind of dyslipidemia unrealted to lipid???

2) Table 5. What the numbers listed in this table stand for?

3) Lines 263-264, edit the size of writing

4) Line 200, edit type error

5) The lack of evaluation of the effects on QOL of drugs such as SGLT-2 inhibitors and GLP-1 RA deserve more space in conclusion. In particular, cite and discuss the effect of IDegLira on QOL of old diabetic patients (doi: 10.1016/j.biopha.2021.112341)

Author Response

Reviewer 3

Thank you so much for your valuable comments. We have responded for your comments point by point as follows. 

In this manuscript the authors examine the difference in health-related quality of life and diabetes-related healthcare events in a large group of adults with diabetes taking a monotherapy including Metformin, Sulfonylurea, Insulin, or Thiazolidinediones (TZD). They find that antidiabetic medications modestly improved QOL during the follow-up period, suggesting that the selection of anti-diabetes medications should be done taking into account also QOL of diabetic patients. The study is interesting. However, it deserves to be deeply improved.

Major concerns:

  • Authors appear to be particularly careful and enthusiasts to provide a long list of acronyms. They occupy larger space of the manuscript for the description of several acronyms rather than try to explain the main findings of this study. Actually, throughout the text this reviewer counts more than 20 acronyms that are really too many. In particular, to read and understand "data source and data collection" as well as "minimun clinically important difference in HRQOL" is difficult and annoying. This part of the manuscript urgently needs to be simplify.

The number of abbreviations were reduced. Unnecessary abbreviations that appear a few times in the text were removed. For instance, (PROs): Patient-reported outcomes, Standard Error of Measurement (SEM), Cohens’ Effect Size (ES), Minimal Detectable Change (MDC) and interquartile range (IQR), home health events (HHE), outpatient events (OP), inpatient events (IPE), emergency room events (ERV).

We made clarifications to the "data source and data collection" as well as "minimum clinically important difference in HRQOL” sections in order to increase the understanding of readers.

2) When does this survey begin? When does it end?

MEPS started in 1996, but our study includes data from 2000 to 2019 since SF-12 administration began in 2000. A complete record of MEPS data has been available since 2019.

3) Please clarify what authors indicate for "home health events"; falls, household accidents, burns or whatelse? What outpatients events include?, hyperglicemia?, hypertensive peaks?; and what includes inpatients events? stroke, myocardial infarction, or nosocomial infection??

The "home health events" variable in MEPS encompasses data on the use of in-home health services, including home health visits, hospice care, and other medical treatments provided in a person's residence. This variable provides insights into the type, frequency, and cost of home health services received by individuals, enabling the analysis of trends and patterns in the utilization and spending of home health services.

Inpatient events and hospitalizations could include diabetic related complications such as diabetic ketoacidosis, myocardial infarctions, stroke, foot, or urinary tract infections. All other events that are related to diabetes but do not require admission to the hospital are categorized under outpatient visits such as kidney, cholesterol, eye exams, A1C test. The reason for all vents and visits should be due to diabetes condition.  

4) In introduction, authors must better explain why they opted to explore the effects of 4 classes of diabetic drugs belonging to an old scientific era. Actually, the recent European and American guidelines for treatment of diabetes indicate SGLT-2-i and GLP1-AR as the first line classes to administer to a diabetic patient.

The American Diabetes Association (ADA) recommends the study medications as the first-line anti-diabetes medications owing to their affordability and proven safety [6]. Previous studies in the US also reported that the use of metformin and insulin analogues have still increased from 1999 to 2018 which derives an ongoing safety and effectiveness study on these agents [7, 8]. Even though the introduction of newer glucose-lowering agents such as GLP-1 agonists and SGLT2 inhibitors improves the management of DM, the newer pharmacotherapeutic options are usually prescribed as a second-line therapy in the case of uncontrolled diabetes. In addition, the accessibility and affordability of these medications limit their utilization for patients mainly with cardio-renal complications. Despite extensive research on the glycaemic effects of these first-line medications, their impact on HRQOL has yet to be explored in a real-world scenario.

5) In Discussion, authors must better distinguish between the different effect of anti-diabetic drugs on outcomes. TZDs (such as pioglitazone) have evident metabolic (doi: 10.1016/j.atherosclerosis.2010.12.021) and cardioprotective effects in patients in secondary CV prevention (PROACTIVE study). Please include these 2 studies. Furthermore, please edit lines 231-233 and lines 236-237. Moreover, the controversial effect of sulfonylureas on CAD must be better discussed

We added the following text in the discussion section of the manuscript: For instance, TZD was reported to confer cardioprotective effect. According to recent studies, TZD such as pioglitazone significantly improves endothelial and adipose tissue dysfunction and reduces the composite of non-fatal myocardial infarction and stroke in patients with type 2 diabetes which might contribute to HRQOL improvement [23, 24]. However, there is a controversy about the role of TZD on CVD outcomes as they were once labelled as a black box warning for worsening of heart failure in diabetes patients [25]. On the other hand, evidence on the effects of sulfonylureas on the heart is still not conclusive and remains as an ongoing debate [26].

Minor concerns

  • Lipid dyslipidaemia? do authors by any chance know any kind of dyslipidaemia unrealted to lipid???

“lipid dyslipidemia” was changed to dyslipidaemia in line 130, under the results section.

  • Table 5. What the numbers listed in this table stand for?

Table 5 indicates the Mean Number of Health Care Associated Events in Diabetes Patients Based on Treatments.

  • Lines 263-264, edit the size of writing

The font size of the sentence was corrected.

  • Line 200, edit type error

The spelling error “Accordning” was to “according” and “indviduals” was changed to “individuals”.

5) The lack of evaluation of the effects on QOL of drugs such as SGLT-2 inhibitors and GLP-1 RA deserve more space in conclusion. In particular, cite and discuss the effect of IDegLira on QOL of old diabetic patients (doi: 10.1016/j.biopha.2021.112341)

We referenced this article in our study which discussed regimen simplification/deprescribing in older diabetes patients and its impact on HRQOL.

 “The modest improvement in HRQOL might also be ascribed to the use of a single blood glucose lowering agent. Rizza S et al 2021 noted that using a single dose of anti-diabetes medication resulted in a greater likelihood of improving health and quality of life as compared to combination therapies [31]”.

In general, future study is warranted to explore the effectiveness of SGLT-2 inhibitors and GLP-1 RA on HRQOL as there is insufficient contemporary real-world data related to these newer agents on HRQOL.  

Round 2

Reviewer 1 Report

Thank you for addressing all my comments. 

Reviewer 3 Report

No more request